# HPLC-Based Metabolomic Analysis and Characterization of *Amaranthus cruentus* Leaf and Inflorescence Extracts for Their Antidiabetic and Antihypertensive Potential

**DOI:** 10.3390/molecules29092003

**Published:** 2024-04-26

**Authors:** Jesús Alfredo Araujo-León, Ivonne Sánchez-del Pino, Rolffy Ortiz-Andrade, Sergio Hidalgo-Figueroa, Areli Carrera-Lanestosa, Ligia Guadalupe Brito-Argáez, Avel González-Sánchez, Germán Giácoman-Vallejos, Oswaldo Hernández-Abreu, Sergio R. Peraza-Sánchez, Andrés Xingú-López, Víctor Aguilar-Hernández

**Affiliations:** 1Unidad de Biología Integrativa, Centro de Investigación Científica de Yucatán (CICY), Mérida 97205, Yucatán, Mexico; jalfredoaraujo@gmail.com (J.A.A.-L.); lbrito@cicy.mx (L.G.B.-A.); 2Unidad de Recursos Naturales, Centro de Investigación Científica de Yucatán (CICY), Mérida 97205, Yucatán, Mexico; isanchez@cicy.mx; 3Facultad de Química, Universidad Autónoma de Yucatán (UADY), Mérida 97069, Yucatán, Mexico; rolffy@correo.uady.mx; 4CONAHCyT-División de Biología Molecular, Instituto Potosino de Investigación Científica y Tecnológica A.C., San Luis Potosí 78216, San Luis Potosí, Mexico; sergio.hidalgo@ipicyt.edu.mx; 5División Académica de Ciencias Agropecuarias, Universidad Juárez Autónoma de Tabasco (UJAT), Villahermosa 86280, Tabasco, Mexico; areli.carrera@ujat.mx; 6Facultad de Ingeniería, Universidad Autónoma de Yucatán (UADY), Mérida 97203, Yucatán, Mexico; avel.gonzalez@correo.uady.mx (A.G.-S.); giacoman@correo.uady.mx (G.G.-V.); 7Centro de Investigación de Ciencia y Tecnología Aplicada de Tabasco, Universidad Juárez Autónoma de Tabasco (UJAT), Cunduacán 86690, Tabasco, Mexico; oswaldo.hernandez@ujat.mx; 8Unidad de Biotecnología, Centro de Investigación Científica de Yucatán (CICY), Mérida 97205, Yucatán, Mexico; speraza@cicy.mx (S.R.P.-S.); andresxl2000@yahoo.com.mx (A.X.-L.)

**Keywords:** betalains, HPLC-UV-DAD, flavonoids, antioxidants, diabetes and hypertension management

## Abstract

The aim of this study was to investigate the potential of *Amaranthus cruentus* flavonoids (quercetin, kaempferol, catechin, hesperetin, naringenin, hesperidin, and naringin), cinnamic acid derivatives (*p*-coumaric acid, ferulic acid, and caffeic acid), and benzoic acids (vanillic acid and 4-hydroxybenzoic acid) as antioxidants, antidiabetic, and antihypertensive agents. An analytical method for simultaneous quantification of flavonoids, cinnamic acid derivatives, and benzoic acids for metabolomic analysis of leaves and inflorescences from *A. cruentus* was developed with HPLC-UV-DAD. Evaluation of linearity, limit of detection, limit of quantitation, precision, and recovery was used to validate the analytical method developed. Maximum total flavonoids contents (5.2 mg/g of lyophilized material) and cinnamic acid derivatives contents (0.6 mg/g of lyophilized material) were found in leaves. Using UV-Vis spectrophotometry, the maximum total betacyanin contents (74.4 mg/g of lyophilized material) and betaxanthin contents (31 mg/g of lyophilized material) were found in inflorescences. The leaf extract showed the highest activity in removing DPPH radicals. In vitro antidiabetic activity of extracts was performed with pancreatic α-glucosidase and intestinal α-amylase, and compared to acarbose. Both extracts exhibited a reduction in enzyme activity from 57 to 74%. Furthermore, the in vivo tests on normoglycemic murine models showed improved glucose homeostasis after sucrose load, which was significantly different from the control. In vitro antihypertensive activity of extracts was performed with angiotensin-converting enzyme and contrasted to captopril; both extracts exhibited a reduction of enzyme activity from 53 to 58%. The leaf extract induced a 45% relaxation in an ex vivo aorta model. In the molecular docking analysis, isoamaranthin and isogomphrenin-I showed predictive binding affinity for α-glucosidases (human maltase-glucoamylase and human sucrase-isomaltase), while catechin displayed binding affinity for human angiotensin-converting enzyme. The data from this study highlights the potential of *A. cruentus* as a functional food.

## 1. Introduction

Over the past decade, amaranth has become an important subject of research because of its high nutritional value and potential as a pseudocereal [1]. *Amaranthus cruentus* L., also known as red amaranth, is one of the many species that has received attention for its exceptional nutritional profile, which includes gluten-free, low calories, and low fat per serving, with high-quality proteins and polyphenols, and for the amaranth-based meals that incorporate the leaves in the traditional medicine diet [2,3].

The high lysine content of amaranth grain distinguishes it from conventional food sources, such as soy, wheat, and rice, providing essential amino acids, high-quality protein, and mineral content that are crucial role in global nutrition [1,4,5,6,7]. Besides its well-balanced amino acid profile and mineral content [7,8], amaranth has a distinct C4 photosynthetic metabolism that provides advantages in dry-matter yield and environmental stress adaptation [2,6]. 

Another distinctive feature of amaranth is the accumulated natural compounds, such as betalains, which give the plant-colored inflorescences, leaves, and stems. The main betalains in amaranth are amaranthin and its epimer isoamaranthin [9]. However, until very recently, the diversity of betalains has become evident [10]. Bioactive compounds such as polyphenols, hydroxycinnamic acids, hydroxybenzoic acids, flavonoids, tannins, and lignins were also reported in amaranth. These compounds have been identified as the cause of the potential antioxidant properties and various pharmacological activities, including antidiabetic, antibacterial or antihelminthic, associated with amaranth that involve reducing oxidative stress [11,12]. Most studies on polyphenols and their impact on antioxidant activity have been focused on *Amaranthus* spp., not *A. cruentus* [13]. The link between betalains and pharmacological activities such as antioxidant, antibacterial, and anti-inflammatory has been reported in species of the *Amaranthus* genus such as *Chenopodium formosanum* and *Gomphrena globosa* [14,15,16].

Amaranth has been used for more than 6000 years since the Aztec civilization attributed medicinal and magical properties to it [17]. In Mexico, *Amaranthus hypochondriacus* seeds are still used for making tamales and a confection of popped seeds with honey called ‘alegrias’ [18]. In Ayurveda and Unani medicine, the amaranth plant has multiple uses. The seeds are spermogenic and tonic. The decoction of the plant is used to reduce the intensity of menstrual bleeding, and the flowers are considered remedies for diarrhea, dysentery, and cough [19,20,21,22,23,24]. In Pakistan, the cooked leaves of *A. cruentus* are utilized as a laxative [25]. Despite its traditional significance, detailed studies that reveal its specific pharmacological properties and phytochemical profile, particularly for metabolic diseases like diabetes and hypertension, are still scarce [26]. Much of the information about the pharmacological properties of Amaranth is based on the analysis of protein hydrolysates, which has revealed bioactive peptides that act as antidiabetics, antihypertensive, and antihypercholesterolemic [27,28,29,30].

Addressing this gap, our study examined the aqueous methanolic extract of *A. cruentus* to determine its influence on critical enzymes related to diabetes and hypertension and its antioxidant capacity. HPLC-UV-DAD and molecular docking were utilized to identify the predominant compounds and elucidate their interactions with target enzymes.

## 2. Results and Discussion

### 2.1. HPLC Chromatographic Conditions for Polyphenols

In this work, an HPLC method was developed and validated for the identification and quantification of flavonoids, cinnamic acid derivates, and the organic acids in *A. cruentus*. Quercetin, kaempferol, catechin, hesperetin, naringenin, hesperidin, and naringin were the flavonoids that were examined. Cinnamic acids were analyzed, including *p*-coumaric acid (pCA), ferulic acid (FA), and caffeic acid (CA). Benzoic acids included vanillic acid (VA) and 4-hydroxybenzoic acid (4-HBA). A reverse-phase chromatography system, which consisted of a C18 column and ternary gradient elution utilizing methanol, acetonitrile, and water supplemented with 0.1% acetic acid, was the most effective method for separating the analytical standards studied. The chromatogram for the phytochemicals was characterized using the absorbance at wavelength of 230 nm. In most cases, the 12 phytochemicals were separated well, as shown by the HPLC chromatogram (Figure 1). Cinnamic acids and catechin that is a polar flavonoid with multiple hydroxyl groups were detected at intervals of 8 to 12 min. Also, glycosylated flavonoids such as naringin and hesperidin were detected. Finally, between 14 and 21 min, the flavonoids aglycones, which do not contain glycosides in their chemical structure, such as quercetin, naringenin, hesperetin, and kaempferol, were eluted.

### 2.2. Validation of Developed HPLC Analytical Method

To validate the developed HPLC method, standard calibration curves were generated in the concentration range of 0.5–5.0 µg/mL and tested parameters such as regression equation parameters, limit of detection (LOD), limit of quantitation (LOQ), and instrumental precision for each analyte. According to the analysis results, each phytochemical had a correlation coefficient exceeding 0.991, indicating good linearity in the calibration curves (Table 1). The formula given in Section 3.5.2 was used to calculate the LOD and LOQ, which were ranging from 0.01–0.10 µg/mL and 0.02–0.34 µg/mL, respectively. Instrumental precision determination was achieved by replicating intraday and interday calibration curves, which showed a relative standard deviation (RSD) of less than 2%. The validation results show that the quality control for the phytochemical analyzed is acceptable and meets the requirements of ICH-Q2A for validating analytical methods.

To evaluate the precision and recovery of the method, 12 standard phytochemicals were tested at different concentrations of 0.5, 2.5, and 5.0 μg/mL through a triple analysis. Intraday and interday measurements were employed to determine both precision and recovery, which were expressed as the percentage of RSD and recovery for each phytochemical (Table 2). The RSD values showed intraday precision between 1.32% and 4.98%, and in interday precision between 1.34% and 4.94%. The percentages of recovery intraday ranged from 83% to 98%, while the percentage of recovery for interday varied between 81% and 98%. The precision and recovery results indicate that the method developed is appropriate.

### 2.3. Quantification of the Polyphenols in A. cruentus

The HPLC analytical method was used to measure simultaneously all 12 phytochemicals, including flavonoids, benzoic acids, and cinnamic acids, in *A. cruentus*. UV-Vis spectra of each phytochemical were recorded and plotted at 230 nm. To confirm each phytochemical peak in the lyophilized *A. cruentus* samples, we compared UV-Vis spectra and retention time with the standard analyte (Figure 2). Quercetin, kaempferol, catechin, and hesperetin were the most abundant flavonoids (Figure 3A), with quercetin, a flavonoid that is not glycosylated, showing the highest concentration at 1361 µg/g in leaves and 77.28 µg/g in inflorescences. The most abundant cinnamic acid was *p*CA, while VA was the benzoic acid with the greatest amount. However, when the HPLC quantification data were calculated to reflect the total content of flavonoids, benzoic acids, and cinnamic acids, we found that the leaves of *A. cruentus* produced significantly more flavonoids (*p* < 0.05) than the inflorescences, with an estimated ratio of 16:1 (Figure 3B). The leaves had a total flavonoid concentration of 5230 ± 403 µg/g compared to the inflorescences, which showed 334 ± 64 µg/g. The total content of benzoic acids and cinnamic acids were also higher in leaves than inflorescences, but the ratio was less than 2, and only the differences in cinnamic acids were significant. The leaves and inflorescences had a total of 317 ± 12 µg/g and 276 ± 47 µg/g of benzoic acids, as well as 643 ± 15 µg/g and 462 ± 58 µg/g of cinnamic acids. Leaves and inflorescences differ significantly in their chemical composition, indicating a more complete polyphenol profile, which could have implications for pharmacological research.

### 2.4. Quantification of Betalains in A. cruentus

Due to the presence of large amounts of betalains in plants, which are classified into two large classes, betacyanins and betaxanthins, we decided to quantify these natural colorants using the spectrophotometry method proposed by Cai et al. (1998) [31]. The method utilizes the molar extinction coefficient of amaranthin 5.66 × 10^4^ cm^−1^ mol^−1^ L to determine betacyanin content and betaxanthins content with indicaxanthin 4.81 × 10^4^ cm^−1^ mol^−1^L. As previously found, *A. cruentus* produces amaranthin and isoamaranthin as its primary betacyanins [32]. In the lyophilized samples of *A. cruentus*, the concentration of these compounds was found to differ significantly (*p* < 0.05) between the inflorescences and the leaves. The inflorescences showed a significant increase in betacyanins, reaching a peak of 74,425 ± 934 µg/g, which was significantly higher than the concentration in the leaves of 17,080 ± 582 µg/g, thus having fourfold more betacyanins (Figure 3C). The betaxanthin content in inflorescences was higher than that found in leaves, with 31,010 ± 389 µg/g and 7445 ± 293 µg/g being the respective amounts. These findings highlight an uneven distribution of betalains between the inflorescences and the leaves, emphasizing the exceptional richness of these pigments in *A. cruentus* inflorescences.

The comparative analysis of 21 *Amaranthus* spp. genotypes by Cai et al. (1998) [31] indicated a betalain concentration range of 16 to 47 mg/g DW in *A. cruentus* inflorescences. This range contrasts sharply with the 74.42 mg/g DW concentration observed in our *A. cruentus* accession. This concentration surpasses those found in the other *Amaranthus* species, such as *A. caudatus* (maximum 28 mg/g), *A. hybridus* (17 mg/g), *A. hypochondriacus* and *A. lividus* (both at 19 mg/g), and *A. paniculatus* (30 mg/g). Additionally, in *A. tricolor*, known for its red leaves, concentrations between 24 and 33 mg/g were reported; in our study, the content in *A. cruentus* accession was lower than in *A. tricolor*, with 17.08 mg/g.

Another study, encompassing 37 species spanning 8 genera within the Amaranthaceae family, with 26 belonging to the genus *Amaranthus*, revealed betacyanin concentrations fluctuating between 0.08 and 1.36 mg/g FW. In this research, *A. cruentus* showed the highest content of 1.36 mg/g FW, significantly lower than the 29.65 mg/g FW concentration in our *A. cruentus* accession, thus indicating a much higher concentration in the latter with a ratio of 21:1. This disparity led to the conclusion that cultivated species generally possess higher betacyanin content compared to their wild counterparts [33]. Furthermore, a study evaluating 48 *Amaranthus* accessions in their vegetative stage found *A. cruentus*, represented by 28 of these accessions, to have the highest betalain concentration at 6.55 mg/g DW [34].

In a broader context, research on red cultivars of Djulis (*Chenopodium formosanum*) found a maximum betalain concentration of 3.4 mg/g FW [14]. In 10 cultivars of nopal (*Opuntia* spp.), betalain contents varied from 0.05 to 5.29 mg/g FW, with betaxanthin totals ranging from 0.12 to 2.86 mg/g [35]. Red beet (*Beta vulgaris*) exhibited betalain contents between 10.26 and 17.15 mg/g DW [36]. A recent study on *Gomphrena globosa* callus reported the highest betacyanin content at 0.37 mg/g FW [37]. In nopal fruit peels (*Opuntia ficus-inidca*), dry-weight betacyanin content was 4.51 mg/g, and betaxanthins were 0.67 mg/g [38]. Freeze-dried and powdered beetroot showed a maximum betacyanin concentration of 4.01 mg/g and betaxanthins at 3.42 mg/g [39]; in pitaya pulp (*Stenocereus thuberi*), total betalain content was 2.6 mg/g DW [40]. This comprehensive comparison allows us to assert that *A. cruentus* from Valladolid accessions is exceptionally rich in betalains, making it a notable natural source of these pigments.

The phenolic components, especially hydroxycinnamic acids, hydroxybenzoic acids, and flavonoids, are essential available plant compounds widely studied across *Amaranthus* species. Comparative studies on *A. tricolor* red genotypes and *A. lividus* green genotypes reveal a richness in phenolic acids and flavonoids in the red variants. Specifically, red accessions showed phenolic concentrations ranging from 85 to 312 µg/g DW, higher than green accessions’ range of 71 to 220 µg/g DW. These concentrations predominantly consisted of gallic acid, vanillic acid, salicylic acid, protocatechin, caffeic acid, *p*-coumaric acid, ferulic acid, and *trans*-cinnamic acids. Moreover, flavonoids such as rutin, quercetin, myricetin, and kaempferol were abundant in red accessions [11,41]. This profile aligns with findings from other studies of *Amaranthus* spp., highlighting the prevalence of flavonoids such as quercetin, rutin, catechin, myricetin, kaempferol, and naringenin [12,42].

In *A. hypochondriacus*, *A. caudatus*, and *A. cruentus*, flavonoids were primarily identified as 3-rutinosides of quercetin and kaempferol, with minimal free quercetin presence. Comparative research with other leafy vegetables, including spinach, lettuce, and both green and red *A. tricolor*, revealed that *A. cruentus* exhibits a superior polyphenol profile, marked by concentrations of 162 µg/g DW for phenolic compounds and 125 µg/g for flavonoids, rich in caffeic acid, *p*-coumaric acid, ferulic acid, isoquercetin, rutin, quercetin, and kaempferol [43].

The polyphenolic content in *A. cruentus* seeds has been documented, revealing a total phenolic concentration of 1214 mg/kg. Kaempferol, forming 86% of this content, was the predominant flavonoid, along with organic acids such as vanillic acid, caffeic acid, and *p*-coumaric acid [44]. In *A. hypochondriacus* seed flours, rutin emerged as the primary flavonoid, with concentrations between 4 and 10 µg/g and a polyphenol profile that included 4-hydroxybenzoic acid, syringic acid, vanillic acid, isoquercitrin, and nicotiflorin [45]. Karamac et al. (2019) observed in *A. caudatus* that early vegetative stages had higher phenolic content (33 mg/g) compared to later stages (27 mg/g) [46]. This content increased through developmental stages, peaking at 27 mg/g during flowering. Leaves showed slightly higher polyphenol concentrations, with a metabolic profile of 17 compounds. Rutin was predominant, ranging from 418 to 1169 µg/g, accounting for 95% of the flavonols. Our results corroborate the findings of Kalinova and Dadakova (2009) [47], who noted that rutin and quercetin were most abundant in leaves compared to inflorescences, while seeds had the highest metabolite content. The amount of quercetin in amaranth leaves is influenced by abiotic factors such as UV-B and UV-C radiation [48].

In summary, our study reveals that the red accession of *A. cruentus* leaves has significantly higher concentrations of organic acids and flavonoids than other species and accessions, marking a notable contribution to understanding phenolic compound distribution in *Amaranthus*. Remarkably, this is the first study to report the polyphenolic profile in the inflorescences of this species.

### 2.5. In Vitro, In Vivo, Ex Vivo, and In Silico Pharmacological Screening of A. cruentus

Pharmacological screening of the aqueous methanol extracts from the leaves and inflorescences of *A. cruentus* was performed to determine their scavenging capacity using the 2,2-diphenyl-1-picrylhydrazyl (DPPH) radical scavenging assay and to determine the inhibitory capacity of enzymes responsible for dietary carbohydrate digestion in humans and regulation of blood pressure. This series of in vitro tests included positive controls such as ascorbic acid due to its high antioxidant capacity, acarbose, which inhibits both pancreatic α-amylase and intestinal α-glucosidase, and captopril, which inhibits the angiotensin-converting enzyme (ACE). The inhibition of pancreatic α-amylase and intestinal α-glucosidase is an effective strategy for diabetes management, as it helps lower and stabilize glucose homeostasis [49,50]. Inhibiting ACE is an effective strategy for hypertension management, as it facilitates lowering high blood pressure by relaxing the veins and arteries [51]. To encourage the creation of amaranthus-based functional foods, the purpose of selecting these enzymes was to investigate their inhibition by the methanolic extract and to determine the extent to which the extract explains the inhibition of those enzymes by acarbose and captopril.

#### 2.5.1. *Antioxidant Effect*

The DPPH assay analysis revealed that extracts from the leaves and inflorescences of *A. cruentus* neutralize the DPPH radical by 52% and 46%, respectively, with no significant differences in antioxidant efficacy (*p* > 0.05) (Figure 4). The current emphasis on antioxidant-rich foods is undeniable, given their crucial role in mitigating oxidative damage linked to the pathogenesis of numerous diseases. An increase in reactive oxygen species (ROS) is often responsible for this damage, which is associated with a range of chronic and degenerative diseases, including diabetes, hypertension, cellular inflammation, Parkinson’s disease, HIV/AIDS, and various cancers [52].

Amaranth (*Amaranthus* spp.) has been recognized as a notable source of antioxidants. Amaranth grains contain significant antioxidant molecules, including squalene, vitamin E, β-carotene, and linoleic acid. *Amaranthus cruentus* is distinguished by its high concentration of phenolic compounds and flavonoids. Additionally, its leaves and inflorescences are rich in betalains, such as amaranthin and its epimer isoamaranthin, which are the most abundant [32]. These findings underscore that amaranth is a promising source for natural antioxidant therapies.

*Amaranthus* spp. are known for their significant antioxidant activity, contributing mainly to their medicinal properties [53]. Studies on *A. spinosus*, *A. viridis*, *A. graecizans*, and *A. hybridus* showed excellent activity in stabilizing the DPPH radical using aqueous methanolic leaf extracts. Specifically, the red accessions of *A. gangeticus* containing high betalain quantities have been observed to play a crucial role in antioxidant effects [54], and this has also been reported in *A. lividus* and *A. hypochondriacus* in comparison [41].

A key group of metabolites contributing to the antioxidant effects of *A. cruentus* is its rich betalain composition. In our previous studies on this *A. cruentus* accession, among the 43 betalains, amaranthin, isoamaranthin, gomphrenin-I, and betanin are the most abundant [32]. Betalains, amino acid derivatives of betalamic acid, possess an amino aromatic chemical structure capable of stabilizing free radicals, for instance, DPPH, through electron donation [55]. Amaranthin and isoamaranthin extracted from *A. tricolor* showed DPPH inhibition values of 31.1%, while gomphrenin-I from *Gomphrena globosa* inhibited 74.5%, and betanin from *Beta vulgaris* showed a 50.8% inhibition rate [56,57].

However, *Amaranthus*’ antioxidant effect is also significantly contributed by its polyphenol profile. The structure of polyphenols, such as flavonoids with semiquinone intermediates, can stabilize free radicals by conjugating aromatic rings and -OH groups. Quercetin, its glycoside rutin, and flavanones naringenin and naringin have shown potent antioxidant power through these pathways [58]. Kaempferol has been demonstrated to neutralize ROS in hepatocellular carcinoma cells [59], suggesting its beneficial effects in antitumor and anti-inflammatory assays [60]. With potent antioxidant properties, catechins are modulated through direct mechanisms, like ROS scavenging by metal ion chelation, or indirect mechanisms that stimulate antioxidant enzyme activation, inhibit prooxidant enzymes, and produce phase II detoxification enzymes [61]. Naringenin and hesperidin, along with their aglycones (naringin and hesperetin), are classified as promising alternatives for cancer prevention and adjuvant therapy due to their potent antioxidant effects, neutralizing ROS and aiding in the modulation of epigenetics, estrogen signaling, apoptosis, and inhibition of tumor metastasis invasion [62].

The metabolic richness of *A. cruentus* in betalains, flavonoids, and organic acids allows these metabolites to be credited with antioxidant power, positioning it as a food that can prevent oxidation in chronic consumption and diseases related to oxidative stress.

#### 2.5.2. *In Vitro and In Silico Inhibition of α-Amylase and α-Glucosidase and In Vivo Oral Sucrose Tolerance Test*

Regarding α-amylase enzyme inhibition, the extracts from inflorescences and leaves did not show significant differences (*p* > 0.05). However, the leaf extract demonstrated greater efficacy (69%) in inhibiting this enzyme, exceeding the inflorescence extract by 17% (52%). A similar pattern was observed in α-glucosidase enzyme inhibition, where the inflorescence extract (54%) surpassed the leaf extract by 8% (46%) (Figure 5).

In a study performed in silico of maltase-glucoamylase (α-glucosidase), four compounds, namely betanin, isobetanin, catechin, and isoamaranthin, demonstrated notable affinity, ranging from −8.1 to −9.1 kcal/mol. These values surpass the co-crystallized ligand acarbose’s affinity of −7.9 kcal/mol (Table 3). Isoamaranthin, with an affinity of −9.1 kcal/mol, stood out by blocking the catalytic site cavity of α-amylase, generating unique interactions with Arg202, Thr205, Arg334, and Phe450, while retaining interaction with Tyr299 (Figure 6A).

Regarding sucrase-isomaltase, another α-glucosidase, all evaluated compounds showed affinities between −6.3 and −9.0 kcal/mol, exceeding the −5.9 kcal/mol affinity of the co-crystallized ligand kotalanol (Table 3). Isogomprhenin I was prominent, with an affinity of −9.0 kcal/mol, internalizing into the catalytic site and maintaining interactions with Asp355, Trp435, and Asp571, like kotalanol, in addition to forming new interactions with Leu233, Asp472, Ser631, Asp632, and Tyr634 (Figure 6B).

Finally, in the case of pancreatic α-amylase, six compounds exhibited affinities comparable or superior to acarbose, ranging from −9.0 to −11.9 kcal/mol, with acarbose having an affinity of −9.1 kcal/mol (Table 3). Once again, isoamaranthin excelled with −11.9 kcal/mol, occupying the catalytic site similarly to acarbose and conserving interactions with Trp59, Gln63, Asp195, and Lys200, in addition to a new interaction with Gly306 (Figure 6C).

The in vitro analysis results from the extracts of both plant tissues inspired their evaluation in an in vivo murine model. Given the extracts’ ability to inhibit key enzymes in the hydrolysis of complex carbohydrates and their notable antioxidant potential, a sucrose tolerance test model was chosen. Sucrose, a disaccharide composed of glucose and fructose linked by a glycosidic bond, can be hydrolyzed by the enzymes mentioned above. The experiment involved administering a glucose load (2 g/kg) to the animals and observing their response to the extract. The hypothesis was that if the extracts inhibited α-amylase and α-glucosidase in an in vivo model, the blood glucose level variation should be lower than a negative control (saline solution). To reach the variation in post-administration blood glucose levels following the sucrose load and the extracts, two drugs were used as positive controls: glibenclamide (10 mg/kg) and acarbose (5 mg/kg), known for their ability to reduce blood glucose levels.

Using aqueous methanol extracts of the leaves and inflorescences of *A. cruentus*, the oral sucrose tolerance test showed a decrease in blood glucose variation compared to the control (saline solution), with a statistically significant difference (*p* < 0.05) (Figure 7). These results indicate that the extracts contribute to better recovery of homeostasis in baseline glucose levels. These findings confirm the existence of a biochemical and pharmacological mechanism in the extracts that modulates and improves the recovery of glycemic homeostasis.

When the behavior of the control drugs was observed, it was noted that glibenclamide, which acts on the pancreatic beta cell by modulating insulin secretion, caused a pronounced decrease in glucose levels. Specifically, at two hours post-administration, a glycemia variation of −12% was recorded; at four hours, it was −46%. On the other hand, acarbose, a drug that inhibits the enzymes responsible for the hydrolysis of complex carbohydrates, showed a blood glucose variation of approximately 2% after two hours post-administration. In comparison, the leaf extracts showed a variation of around 5%, the inflorescence extracts around 11%, and the saline solution control showed a variation of 22%.

Enzyme evaluations of α-glucosidase and α-amylase, and the oral sucrose tolerance test, were conducted to demonstrate that extracts from *A. cruentus* might assist in diabetes prevention. Intestinal α-glucosidase inhibitors slow down the absorption rate of complex carbohydrates, reducing postprandial hyperglycemia. Commercial inhibitors (including acarbose, miglitol, and voglibose) act by competitively inhibiting at the border of the enterocytes lining of the intestinal lumen, preventing the absorption of disaccharides such as sucrose and oligosaccharides but not affecting monosaccharides such as glucose [63]. These inhibitors have also been noted to alter the release of glucose-dependent intestinal hormones, increasing glucagon secretion and decreasing postprandial insulin concentrations [64].

Supplementation with 30% amaranth flour (*A. hypochondriacus*) in blue corn tortillas has been reported to inhibit enzymes α-amylase and α-glucosidase [65]. Additionally, aqueous methanolic extracts from amaranth grains (*A. cruentus*) and leaves (*A. hybridus*) consumed in Kenya successfully inhibited α-amylase and α-glucosidase. Grain extracts presented a 35% and 40% inhibition for α-amylase and α-glucosidase, respectively, while leaf extracts showed a 27% and 28% inhibition, respectively. The study concluded that this inhibitory effect was due to phenolic metabolites with antioxidant capacity, such as flavonoids and organic acids [66]. This enzyme inhibition was also observed in aqueous extracts from *Amaranthus* sp. stems, where flavonoids were identified as the primary phenolic compounds responsible [67]. Edible leaves of *A. inamoenus* and *A. gangeticus* from Taiwan displayed this antidiabetic potential by inhibiting the mentioned enzymes linked to polyphenolic compounds [68]. A recent study showed that shallot enriched with amaranth reduced blood glucose levels in diabetic rats induced by streptomycin and treated with chronic doses for 14 days. The modulation of this effect was linked to intestinal α-glucosidase enzymes and the potential of phenolic compounds [69].

The inhibitory activity of polyphenols (organic acids, flavonoids, and catechins) is well documented, with studies investigating how hydroxyl positions affect structure–activity relationships [70]. Our in silico studies found that isoamaranthin and isogomprhenin-I exhibited the lowest ΔG in their molecular interaction with glucosidase and amylase enzymes. While further experimental research will be necessary to study the interactions of isoamaranthin with the enzyme more precisely, one of the immediate challenges might be the isolation of high-purity isoamaranthin. It is established that red beet extract, known for its high betanin content, can inhibit both enzymes in the intestinal lumen, leading to reduced postprandial glucose levels [71]. Aqueous ethanolic extracts from *Celosia argentea* inflorescences, with a betacyanin content of 14.52 mg/g and abundant amaranthin and isoamaranthin, have shown enzyme inhibitory activity [72]. However, the literature on the correlation between betalains and enzyme inhibition is scarce. Some studies suggest that betalains and polyphenols work together in certain species of the Amaranthaceae family to inhibit these enzymes, suggesting a synergistic effect that modulates enzymatic activity [73,74].

During the in vivo study, we observed that both extracts from *A. cruentus* at 200 mg/kg concentrations were markedly more effective in improving glucose homeostasis than the saline solution in the control group. Their behavior differed meaningfully from the kinetics of glibenclamide, a drug that stimulates insulin secretion, signifying that the extracts likely do not possess secretagogue activity. Amaranth extracts may affect the digestion of carbohydrates, which is in line with the observed inhibition of pancreatic α-amylase and intestinal α-glucosidase, and with the similar glycemic index pattern caused by extracts or acarbose administration. Integrating the metabolomic profile, in vitro activity, and in silico molecular dynamics, it is possible that betalains are the metabolites responsible for this activity, and they interact synergistically with the present group of polyphenols. The oral bioavailability of betalains from red beet (*Beta vulgaris*) has been studied in murine models [74], as well as the betalains from prickly pear fruit (*Opuntia ficus-indica*) in acute inflammation rodent models [75]. Furthermore, the antidiabetic effects of betanin in streptozotocin-induced rats have been observed, revealing that betanin’s effects are mediated through the adenosine 5′-monophosphate-activated protein kinase (AMPK)/silent information regulator 1 (SIRT1)/nuclear factor kappa-light-chain-enhancer of activated B cells (NF-κB) signaling pathway, positioning betalains as effective phytochemicals in treating metabolic disorders like diabetes [76].

#### 2.5.3. *In Vitro and In Silico Inhibition of Angiotensin-Converting Enzyme and Ex Vivo Vasorelaxation Experiment*

Last, both extracts showed comparable efficacy in inhibiting the ACE, with the leaf extract inhibiting by 58% and the inflorescence extract inhibiting by 53%, with only a 5% difference (Figure 8A). Since *A. cruentus* extracts showed significant activity (>50%) against ACE, we explored their potential effect on murine aortic vasorelaxation through an ex vivo experiment. We observed that the leaf extract induced a maximum vasorelaxation of approximately 45% in endothelium-intact aortic rings (E+), outperforming the inflorescence extract, which achieved a rate of 28% under similar conditions. These effects were endothelium-specific, as aortic rings without endothelium (E−) did not exhibit significant changes in vasorelaxation (Figure 8B).

Research on the effects of amaranth on ACE inhibition is scarce. Studies highlight that polar extracts from amaranths, specifically *A. hybridus* and *A. dubius* cultivated in KwaZulu-Natal, South Africa, have shown promising ACE inhibitory activity [77]. *Amaranthus hypochondriacus* has a significant seed component, *11S* globulin, which is known for inhibiting ACE, and is where antihypertensive tetrapeptides can be found [78]. Further investigations into this species have led to the isolation of ACE-inhibiting peptide fractions from albumin 1 and the grain globulin. Notably, albumin 1 and globulin exhibited the highest ACE inhibitory activities of 40% and 35% after hydrolysis periods of 18 and 15 h, respectively [79]. The efficacy of its *7S* globulin in inhibiting ACE has also been confirmed [80,81]. Moreover, amaranth protein hydrolysates prepared with lactobacillus strains (*L. plantarum* and *L. helveticus*) have demonstrated notable ACE inhibition, with effectiveness reaching 45.9% [82].

For the in silico approach in ACE, all evaluated compounds, except for isoamarantin, demonstrated affinity within the range of −8.0 to −9.9 kcal/mol, surpassing the affinity of the co-crystallized ligand lisinopril, which had an affinity of −7.6 kcal/mol (Table 1). Notably, catechin, with an affinity of −9.9 kcal/mol, exhibited the highest affinity for this enzyme. Catechin occupies a portion of the more exposed cavity and retains interactions with lisinopril’s binding residues, including His353, Ala354, Val380, and Tyr523 (Figure 6D).

Betalains from fresh red beetroot juice have been shown to exhibit inhibitory activity against ACE, with up to 87% inhibition [83]. Similarly, polyphenolic compounds are described as potent ACE inhibitors due to their antioxidant capacity [84,85]. To date, no study in the literature has described amaranth’s comprehensive metabolic profile, particularly linking ACE inhibition to betalains and catechins found in the leaves and the inflorescences of *A. cruentus*. Moreover, while vasorelaxation in the murine aorta may not have clinical significance, it is noteworthy that amaranth extract contains metabolites that induce mild vasorelaxation. This finding paves the way for chronic dosage studies of *A. cruentus* extract to assess aortic reactivity, as prolonged exposure to this extract might activate mechanisms involving endothelial nitric oxide (NO). In the vascular endothelium, L-arginine is utilized by endothelial NO synthase (eNOS) to produce NO and to promote guanosine 3′,5′-cyclic monophosphate (cGMP) formation. Subsequently, cGMP leads to vasodilation, a crucial process in maintaining vascular homeostasis and regulating vascular tone [86,87].

## 3. Materials and Methods

### 3.1. Plant Material

The seeds of red amaranth (*A. cruentus*) were collected in Valladolid, Yucatán, Mexico. The collected materials were identified by Ivonne Sánchez del Pino (Ph.D.). *Amaranthus cruentus* plants were cultivated in the open-air gardens and experimental greenhouses at CICY. Leaves and inflorescences were harvested from the plants after 60 days of cultivation. Once collected in the field, the tissues were frozen in liquid nitrogen and freeze-dried at −51 °C for 72 h. Once dried, the tissues were ground using a blade mill and stored in the dark at −20 °C until analysis.

### 3.2. Chemical and Reagents

Methanol (MeOH) and water (H_2_O) of HPLC grade (Tedia, Fairfield, CT, USA) were used as extraction solvents. For HPLC-UV-DAD analysis, MeOH, acetonitrile (ACN), and H_2_O of HPLC grade (Tedia, Fairfield, CT, USA) were used, with the mobile phase acidified using acetic acid of HPLC grade (Fisher Scientific, Waltham, MA, USA). A beetroot extract (Sigma-Aldrich, Cat. # 901266, St. Louis, MO, USA) was used as a reference material. For matrix solid-phase dispersion (MSPD) extraction, BondElut-C18 was used as the stationary phase, acquired from Agilent Technologies with a particle size of 40 µm (Santa Clara, CA, USA).

For the HPLC-UV-DAD quantification of flavonoids and organic acids, the following analytical standards were acquired from Sigma-Aldrich with a purity exceeding 99%: hesperidin, hesperetin, naringin, naringenin, quercetin, catechin, kaempferol, vanillic acid, 4-hydroxy-benzoic acid, *p*-coumaric acid, caffeic acid, and ferulic acid. For betalains, betanin from a beetroot extract was used.

### 3.3. Phytochemical Extraction from Leaves and Inflorescences of A. cruentus

Phytochemicals, including betalains, were extracted via the matrix solid-phase dispersion (MSPD) method published by Araujo-León et al. (2023) [32]. A uniform powder was obtained by grinding 100 mg of lyophilized tissue at room temperature in a mortar with 400 mg of C18 stationary phase (BondElut-C18). The solid mixture was placed into cartridges for solid-phase extraction. With the aid of a vacuum manifold (Visiprep, SUPELCO, Bellefonte, PA, USA), 9 mL of 0.1% acetic acid-water was eluted, followed by a mixture of 0.1% acetic acid-water:methanol (1:1, *v*/*v*) mixture at −15 mmHg. Both eluates were collected and evaporated to dryness. The resulting extract was resuspended in 0.1% acetic acid-water:methanol (1:1, *v*/*v*) mixture and transferred to an amber vial for analysis by HPLC-UV-DAD.

### 3.4. HPLC-UV-DAD Analysis

The HPLC-UV-DAD analyses were conducted using an Ultimate 3000 UHPLC system (Thermo Scientific, Waltham, MA, USA) coupled with a UV/Vis detector (Ultimate 3000 UV/VIS detector, Dionex Co., San Jose, CA, USA). A 20 µL aliquot of the sample was injected, transferred to a stainless-steel loop, and then to a Hypersil GOLD C18 column (100 × 2.1 mm, 1.9 µm particle size, Thermo Scientific, San Jose, CA, USA) at a flow rate of 300 µL/min. The sample was eluted from the column at a flow rate of 350 µL/min using a gradient with 0.1% acetic acid-water (solvent A), 0.1% acetic acid-methanol (solvent B), and 0.1% acetic acid-acetonitrile (solvent C). The column elution system started with 100% solvent A for 2 min. From minutes 2 to 16, the proportion of organic solvent was increased to 80% solvent B and 20% solvent C. Finally, the column was purged for 8 min and stabilized to the initial run proportion. UV/Vis spectra data were collected and plotted at 230 nm. Peak areas were determined using Chromeleon 7.1 software (Thermo Scientific, Waltham, MA, USA). Graphs were generated using GraphPad Prism version 9.0 (GraphPad, San Diego, CA, USA).

### 3.5. HPLC-UV-DAD Method Validation

The method was validated according to the criteria established in the ICH Q2A guideline, evaluating precision, accuracy, linearity, range, and the limits of detection and quantification.

#### 3.5.1. Linearity

The linearity test was conducted both intraday (*n* = 3) and interday (*n* = 9). Five multicomponent solutions containing all analytical standards were prepared in a range of 0.5–5.0 µg/mL and analyzed by HPLC-UV-DAD. The linear regression equation was generated using the least squares method, considering the area under the curve of the analyte versus concentration.

The determination of the slope (*b*), intercept (*a*), correlation coefficient (*r*), and residual variance $(S_{y,x}^{2}$) was performed using the following formulas (ICH Q2A):$$b=\frac{\sum xy-\frac{\sum x\sum y}{n}}{\sum x^{2}-\frac{{\left(\sum x\right)}^{2}}{n}}$$
$$a=\frac{\sum y-b\sum x}{n}$$
$$r=\frac{\sum \left(x-\left.x_{m}\right)\left(y-y_{m}\right)\right.}{\sqrt{\sum {\left(x-x_{m}\right)}^{2}\sum {\left(y-y_{m}\right)}^{2}}}$$
$$s_{y,x}^{2}=\frac{\sum {\left(y-y_{m}\right)}^{2}}{n-2}\left(1-r^{2}\right)$$
where *x* is the independent variable, *x_m_* the overall mean of the independent variable, *y* is the dependent variable, *y_m_* is the overall mean of the dependent variable, and *n* is the total number of trials conducted.

#### 3.5.2. Limits of Detection and Quantification

The detection and quantification limits were determined by a test using solutions in the range of 80 to 120%. A calibration curve with the lowest point (0.5 µg/mL) evaluated for instrumental linearity was constructed for this.

To calculate the limits, the residual standard deviation (*S_y/x_*) was used, and the following formula was established (ICH Q2A):$$LOD=\frac{3S_{y/x}}{b}$$
$$LOQ=\frac{10S_{y/x}}{b}$$
where *S_y/x_* = residual standard deviation or noise, *b* = slope, *LOD* = limit of detection, and *LOQ* = limit of quantification.

#### 3.5.3. Precision and Accuracy

The precision assessment was conducted intraday (*n* = 3) and interday (*n* = 9), at three calibration curve points (0.5, 2.5, and 5.0 µg/mL). The area under the curve (AUC) of each analyte was used to calculate the relative standard deviation (RSD) and evaluate the percentage of deviation concerning the mean for each assay.

Accuracy was determined through standard addition-recovery (ICH Q2A), assessing 9 determinations at different concentrations (0.5, 2.5, and 5.0 µg/mL). With the data obtained from the area under the curve, the percentage of bias for intraday and interday was calculated, evaluating the percentage of deviation concerning the mean for each assay.

### 3.6. Betalains and Betaxanthins Quantification by Spectrophotometry

The content of betalains and betaxanthins was determined using the spectrometric method proposed by Cai et al. (1998) [31]. A total of 100 mg of dry tissue (leaves and inflorescences) was extracted with an H_2_O:MeOH (80:20, *v*/*v*) mixture acidified with 0.1% acetic acid. The plant material was placed in a 50 mL centrifuge tube, and 25 mL of the extraction mixture was added. The mixture was vortexed for 10 min and centrifuged at 9000 rpm at 20 °C. The supernatant was transferred to amber vials for spectrophotometric analysis.

Betacyanins and betaxanthins quantification was determined using the following formulas:Betacyanins content (mg/g) = (A_536_) (MW)(V)(DF)/εLW
Betaxanthins content (mg/g) = (A_480_) (MW)(V)(DF)/εLW

A_536_ represents the absorbance measured at 536 nm for betacyanin content, MW is the molecular weight of amaranthin (726.6 g/mol), and ε is the molar extinction coefficient for amaranthin (5.66 × 10^4^ cm^−1^ mol^−1^ L). A_480_ represents the absorbance measured at 480 nm for betaxanthin content, MW is the molecular weight of indicaxanthin (308 g/mol), and ε is the molar extinction coefficient for indicaxanthin (4.81 × 10^4^ cm^−1^ mol^−1^ L). V is the total volume of the extract in mL, DF is the dilution factor, and W is the dry weight of the plant tissue used in grams.

### 3.7. Phenolic Compounds Quantification

Appropriate analytical standards were used to determine the concentrations of organic acids (vanillic acid, 4-hydroxybenzoic acid, *p*-coumaric acid, caffeic acid, and ferulic acid) and flavonoids (hesperidin, hesperetin, naringin, naringenin, quercetin, catechin, and kaempferol). Calibration curves were constructed using an external standard of 0.5 to 5 µg/mL. The determination was carried out using the UV spectral characteristics of each phytochemical, retention times, and co-chromatography with samples fortified with the available standards. Finally, the total contents of organic acids and flavonoids were reported by summing up the individual concentrations of each compound.

### 3.8. DPPH Assay

To evaluate the free radical scavenging capacity of the extracts, the methodology established by Shimada et al. (1992) was followed [88], utilizing the 1,1-diphenyl-2-picrylhydrazyl (DPPH) radical. Initially, the dried aqueous methanolic extracts of leaves and inflorescences of *A. cruentus* were dissolved. Then, aliquots of 150 µL were combined with 1350 µL of DPPH (0.1 mM in ethanol). The mixture was vortexed and allowed to react for 30 min at 25 °C in amber vials to protect it from light. Once the reaction concluded, the absorbance was measured using a spectrophotometer (Thermo Scientific Evolution 220 UV-VIS, Thermo Scientific, Waltham, MA, USA) at a wavelength of 517 nm. Ascorbic acid (vitamin C) was used as a positive control for the reaction due to its high antioxidant capacity, and all determinations were conducted in triplicate.

The percentage of DPPH radical scavenging was calculated using the following formula:$$\%\;Scavenging\;capacity=100*\left[\frac{CA-EA}{CA}\right]$$
where *CA* is the absorbance of the control and *EA* is the absorbance in the presence of the extract.

### 3.9. In Vitro α-Amylase Inhibition Assay

To determine the α-amylase enzyme activity in the aqueous methanol extracts from the leaves and inflorescences of *A. cruentus*, we modified the method described by Tamil et al. (2010) [89]. In 200 µL tubes, 2 mg of starch was suspended in 0.5 M Tris-HCl (pH 6.9) with 0.01 M CaCl_2_. The solution was boiled for 5 min then incubated at 37 °C for 5 min. Subsequently, we added 200 µL of 50% (*v*/*v*) dimethyl sulfoxide, 200 µL of the extracts at a concentration of 1000 µg/mL, 200 µL of porcine pancreatic α-amylase (2 U/mL), and 500 µL of 0.1% 3,5-dinitrosalicylic acid to each tube. This mixture was allowed to react for 10 min at 37 °C, and the reaction was halted by adding 0.5 mL of 50% acetic acid to each tube. The tubes were then centrifuged at 4500 rpm at 4 °C for 10 min, and the supernatant was analyzed using spectrophotometry.

The absorbance of the supernatant was measured at a wavelength of 540 nm. Acarbose was used as a positive control with a level of 100 μg/mL (154.89 μM), achieving an IC50 value of 83.33 μg/mL (129.07 μM) [89]. All analyses were performed in triplicate.

The inhibitory activity was calculated using the following equation:Inhibitory activity = [(Ac+) − (Ac−) − (As − Ab)]/[(Ac+) − (Ac−)] × 100
where Ac+ is the absorbance when the enzyme acts without interference (solvent with enzyme), Ac− is the absorbance when the enzyme is not active (solvent without enzyme), and As is the absorbance when the enzyme acts in the presence of the sample. Ab is the absorbance of the blank.

### 3.10. In Vitro α-Glucosidase Inhibition Assay

For the in vitro α-amylase inhibition assay, 2 U/mL of α-glucosidase was dissolved in phosphate buffer and incubated for 5 min at 37 °C. Following the incubation, 100 μL of *p*-nitrophenyl-glucopyranoside, pre-dissolved in phosphate buffer (50 mM, pH 6.8), was added to initiate the reaction. The mixture was then allowed to react at 37 °C for 1 h. A total of 250 μL of 1.0 M sodium carbonate was added to each tube to halt the reaction. The tubes were centrifuged at 4500 rpm at 4 °C for 10 min, and the supernatant was subjected to spectrophotometric analysis. The absorbance of the resulting mixture was measured at a wavelength of 405 nm. Acarbose was used as a reference compound with a level of 100 μg/mL (154.89 μM), achieving an IC50 value of 83.33 μg/mL (129.07 μM) [89]. All analyses of the aqueous methanol extracts from the leaves and inflorescences of *A. cruentus* were conducted in triplicate at 1000 µg/mL concentration. Inhibitory activity was determined using the proposed equation:Inhibitory activity = [(Ac+) − (Ac−) − (As − Ab)]/[(Ac+) − (Ac−)] × 100
where Ac+ is the absorbance when the enzyme acts without interference (solvent with enzyme), Ac− is the absorbance when the enzyme is inactive (solvent without enzyme), and As is the absorbance when the enzyme acts in the presence of the sample. Ab is the absorbance of the blank.

### 3.11. In Vitro Angiotensin-Converting Enzyme (ACE) Inhibition Assay

The inhibitory activity of ACE against aqueous leaf extracts and inflorescences from *A. cruentus* was assessed at 1 mg/mL. For this test, the Hayakari et al. (1978) method was followed [90], where ACE hydrolyzes hipuril-L-histidil-L-leucine into hippuric acid and His-Leu. This approach relies on the colorimetric reaction of hippuric acid with TT (2,4,6-trichloro-s-triazine) [91]. For the assay, 20 µL of ACE solution (100 mU/mL) was mixed with 40 µL of 1 mg/mL sample extracts and incubated in a water bath (VWR^®^ Heating circulator model 1130-2S, VWR International, Radnor, PA, USA) at 37 °C for 5 min. Subsequently, 100 µL of 0.3% HHL (hippuril-L-histidil-L-leucine) was added to a buffer composed of 40 µmol potassium phosphate and 300 µmol sodium chloride, adjusted to pH 8.3. The mixture was incubated at 37 °C for 45 min. The reaction was halted by adding 360 µL of TT solution in dioxane (3%, *w*/*v*) and 720 µL of 0.2 M phosphate buffer (pH 8.3). The mixture was then centrifuged at 10,000 rpm for 10 min, and the absorbance of the supernatant was measured at 382 nm using a spectrophotometer (Thermo Scientific Evolution 220 UV-VIS, Thermo Scientific, Waltham, MA, USA). Captopril served as the reference standard at 50 μg/mL (230.11 μM), achieving an IC50 value of 28.7 nM [90]. All assays were conducted in triplicate.

### 3.12. In Vivo Sucrose Tolerance Test

Male Wistar rats (250–270 g) were obtained from the animal house of Universidad Juárez Autónoma de Tabasco. The animals were housed under standard laboratory conditions with a 12 h light–dark cycle at a temperature of 25 °C and a humidity level of about 65%. They had access ad libitum to a standard diet and water. All experimental procedures complied with the guidelines of the Mexican Federal Standard for Animal Experimentation and Care (NOM-062-ZOO-1999) and the 2010 Guide for the Care and Use of Laboratory Animals (No. 85-23, revised edition). An ethics committee approved this animal experimentation protocol in October 2017 under certification code 2017-001 (UJAT).

We used normoglycemic rats after a 16 h fasting period for the sucrose tolerance test. The experimental groups consisted of five animals each, and all administrations were conducted via an intragastric route. The first group was a control, receiving only 0.9% (*w*/*v*) of the NaCl saline solution. The second group was a positive control with acarbose at a dose of 5 mg/kg. The third group was another control with glibenclamide at 10 mg/kg. The fourth group received an aqueous methanol extract of *A. cruentus* leaves at 200 mg/kg, and the fifth group received an aqueous methanol extract of *A. cruentus* inflorescences at the same dose. Following the protocol of Araujo-León et al. (2023) [92], glucose measurements were taken after administering a sucrose solution (2 g/kg) at 0, 0.5, 1, 2, and 4 h. Blood glucose levels were measured using a commercial glucometer (Accu-Check Active, Roche, Basel, Switzerland). The glucose variation (% GV) for each group was plotted, considering the initial value at 0 h, according to the following formula:$$\%GV=\frac{G_{x}-G_{0}}{G_{0}}*100$$
where *G*_0_ represents the initial glycemia at time 0 h, and *G_x_* corresponds to the glycemia values at times 0.5, 1, 2, and 4 h, respectively.

### 3.13. Molecular Docking Study

#### 3.13.1. Ligands Preparation

The ligand structures of amaranthin, betanin, catechin, gomphrenin-I, hesperetin, isoamaranthin isobetanin, isogomprhenin-I, kaempferol, naringenin, and quercetin were optimized by using the MMFF94x force field, which is implemented in MOE software (v2022.02; MOE 2022). First, ligands were protonated and calculated partial charges, and the conjugate gradient algorithm was employed, with the RMS gradient set to 0.0001 kcal/mol/Å2.

#### 3.13.2. Target Preparation

The X-ray crystal structures of the human ACE (PDB id: 1O86, resolution of 2 Å), human maltase-glucoamylase (PDB id: 2QMJ, resolution of 1.9 Å), human sucrase-isomaltase (PDB id: 3LPP, resolution of 2.15 Å), and human pancreatic alpha-amylase (PDB id: 3BAJ, resolution of 2.1 Å) were downloaded from the RCSB Database [93]. The protein structures were prepared by removing foreign molecules and water using MOE software.

#### 3.13.3. Target Preparation

Virtual screening was carried out using AutoDock Vina [94,95] with eleven potential inhibitors (amaranthin, betanin, catechin, gomphrenin-I, hesperetin, isoamaranthin, isobetanin, isogomprhenin I, kaempferol, naringenin, quercetin) and the respective co-crystal molecules (lisinopril, acarbose, and kotalanol). The grid box sizes for the human ACE receptor were 24 × 22 × 22, with grid center x = 40.89, grid center y = 32.39, and grid center z = 47.28; for the human maltase-glucoamylase receptor, they were 24 × 22 × 22, with grid center x = −21.70, center y = −6.32, center z = −5.61; for the human sucrase-isomaltase, receptor were 26 × 24 × 24, with grid center x = 38.52, grid center y = 58.56, grid center z = 79.25; for human pancreatic alpha-amylase receptor, they were 40 × 28 × 26, with grid center x = 8.44, grid center y =15.66, grid center z = 40.31; all use a default grid spacing of 1 Å.

## 4. Conclusions

This study evaluated aqueous methanol extracts from the leaves and inflorescences of *A. cruentus* and found that they have significant pharmacological and therapeutic potential, particularly highlighting their benefits for diabetes management. Their ability to inhibit critical enzymes (α-amylase and α-glucosidase) involved in the hydrolysis of complex carbohydrates, and their notable antioxidant action and capacity to inhibit the angiotensin-converting enzyme (ACE) contribute to this effect. These combined mechanisms are crucial for the comprehensive management of metabolic syndrome. The results indicated a significantly higher concentration of betalains in the inflorescences than in the leaves, while the latter exhibited a higher concentration of flavonoids. Based on the findings from in vivo models, the hypothesis that the extracts possess biochemical and pharmacological properties that are beneficial for glycemic homeostasis was supported. When comparing the effects of the extracts with control drugs such as glibenclamide and acarbose, it was found that the extracts have a moderate yet significant impact on blood glucose regulation, suggesting a milder action profile that could be advantageous in various clinical scenarios. With these results obtained in both phytochemical profiling and pharmacological screening, we present the first evidence supporting *Amaranthus cruentus* L. as a functional food. According to the Food and Agriculture Organization (FAO), functional food is a source of extra components that enhance health [96]. This research could enhance the perception of amaranth as a rich resource of phytochemicals, which could support its medicinal validity and lead to new therapeutic applications.

## Figures and Tables

**Figure 1 molecules-29-02003-f001:**
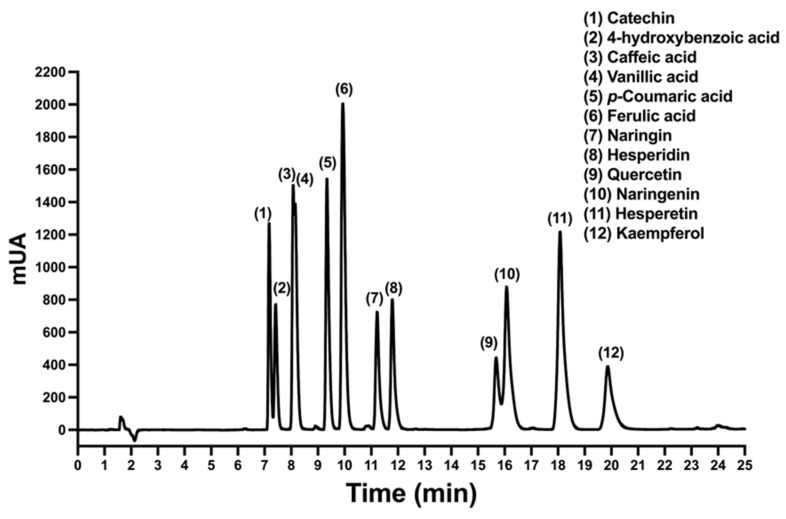
HPLC-DAD chromatogram of the analytical standards for quantifying flavonoids, cinnamic acids and benzoic acids at the wavelength of 230 nm.

**Figure 2 molecules-29-02003-f002:**
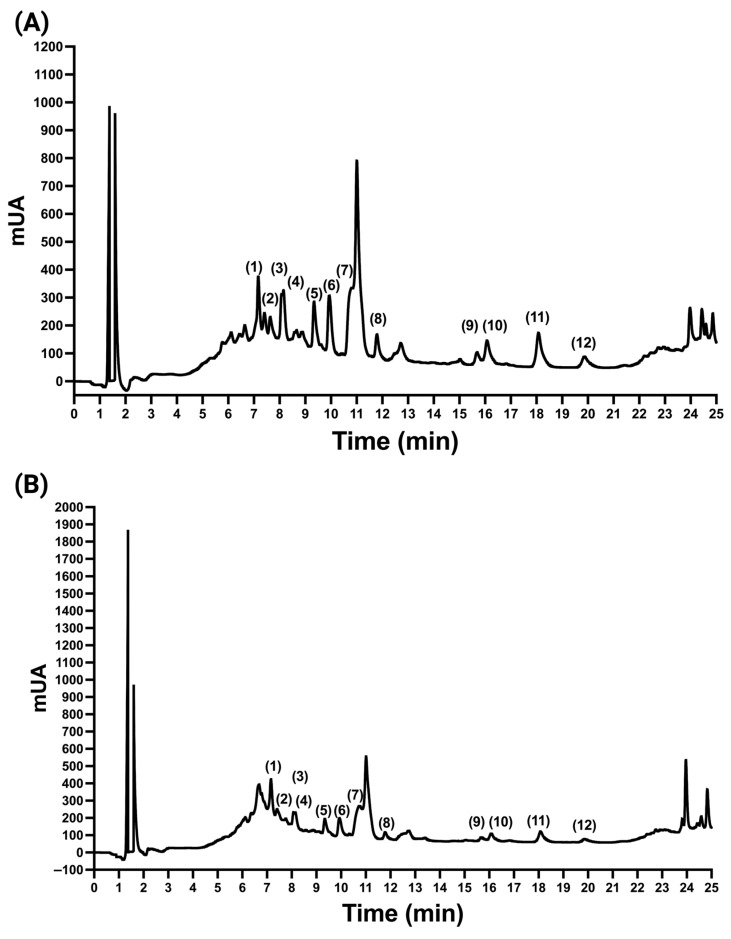
Chromatograms at 230 nm of samples from leaves (**A**) and inflorescences (**B**). The peaks labeled 1–12 correspond to the phytochemicals with the same label in Figure 1.

**Figure 3 molecules-29-02003-f003:**
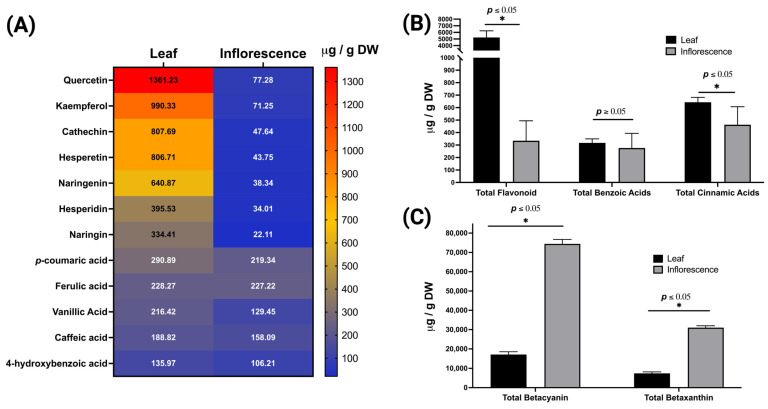
Chemical composition of polyphenols and betalains in leaves and inflorescences of *A. cruentus*. (**A**) Heatmap visualization of polyphenol concentrations; (**B**) total polyphenol content including flavonoids, benzoic acids, and cinnamic acids; (**C**) total betalain content, comprising both betacyanins and betaxanthins.

**Figure 4 molecules-29-02003-f004:**
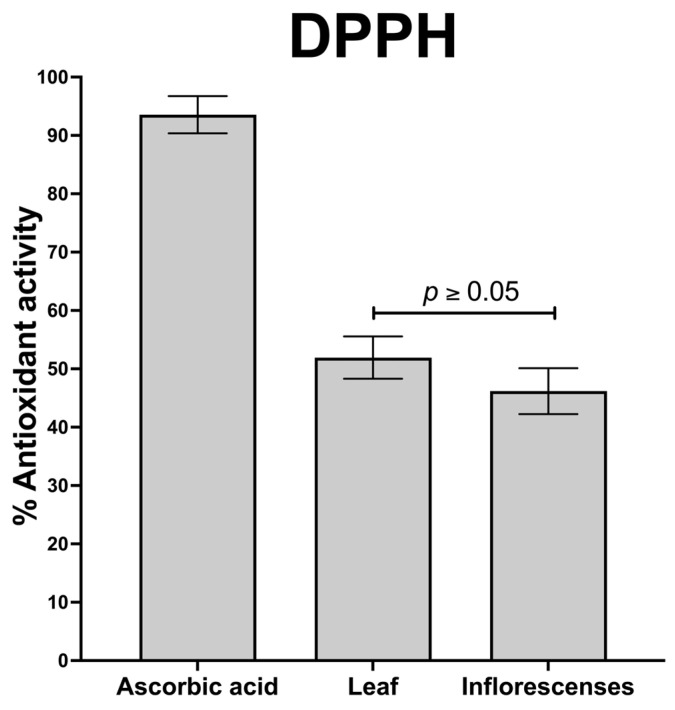
DPPH radical scavenging activities of the leaf and inflorescences of *A. cruentus*.

**Figure 5 molecules-29-02003-f005:**
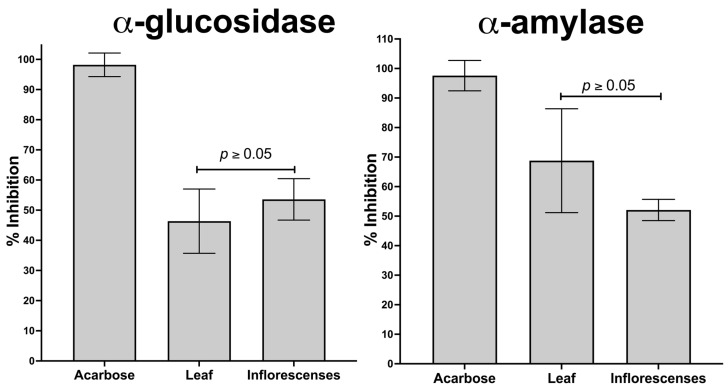
Comparison of inhibitory activity of α-glucosidase and α-amylase in leaf and inflorescence extracts of *A. cruentus*.

**Figure 6 molecules-29-02003-f006:**
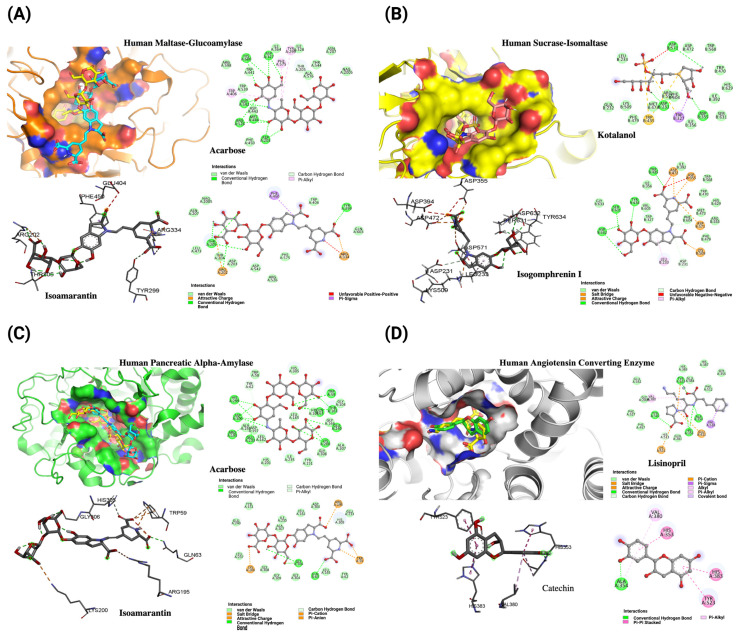
Structural and interaction analysis of key human enzymes with inhibitors, accompanied by 2D interaction diagrams. (**A**) Maltase-glucoamylase bound with acarbose (yellow) and isoamaranthin (−9.1 kcal/mol, cyan). (**B**) Sucrase-isomaltase interaction with kotalanol (yellow) and isogomphrenin-I (−9.0 kcal/mol, salmon). (**C**) Pancreatic alpha-amylase complexed with acarbose (yellow) and isoamaranthin (−11.9 kcal/mol, cyan). (**D**) Angiotensin-converting enzyme complexed with lisinopril (yellow) and catechin (−9.9 kcal/mol, green).

**Figure 7 molecules-29-02003-f007:**
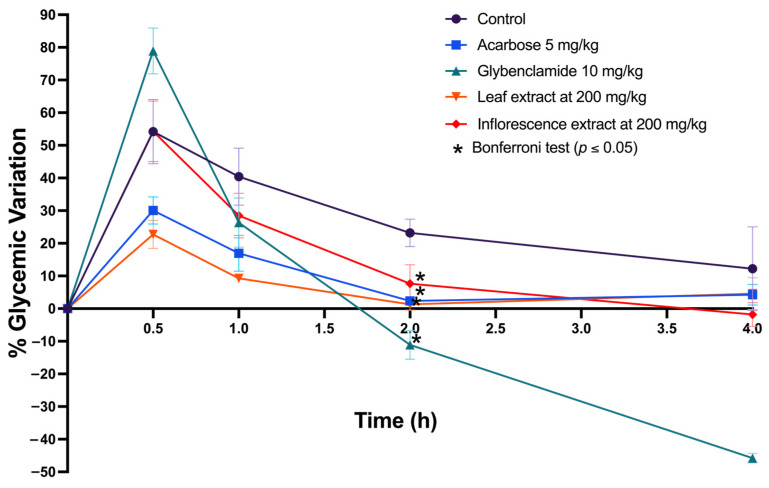
Sucrose tolerance test in a normoglycemic murine model using leaves and inflorescence extracts of *A. cruentus*.

**Figure 8 molecules-29-02003-f008:**
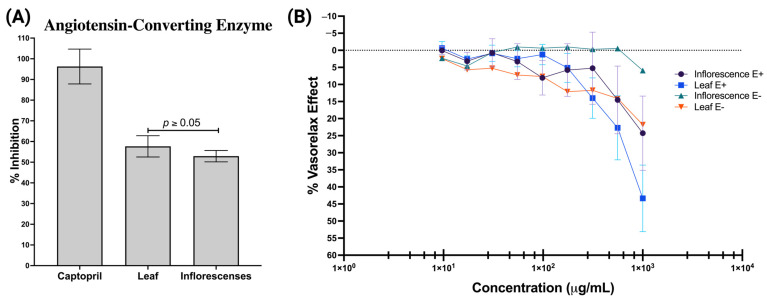
Vasodilator effect of *A. cruentus* extracts from leaves and inflorescences. (**A**) Angiotensin-converting enzyme inhibition and (**B**) concentration–response curves of the vasorelaxant effect: E+: with endothelium and E−: without endothelium.

**Table 1 molecules-29-02003-t001:** Quality control for phytochemicals assessed.

Phytochemical	*t*_R_ (min)	Wavelength (nm)	Slope (b)	Intercept (a)	r	Noise (S_y/x_)	LOD (µg/mL)	LOQ (µg/mL)	Instrumental Precision at 0.5 µg/mL (% RSD)
Catechin	7.15	280	0.45	−0.05	0.9981	0.015	0.10	0.34	0.98
4-HBA	7.41	254	6.41	−0.65	0.9932	0.030	0.01	0.05	1.34
CA	8.11	320	1.29	−0.35	0.9924	0.033	0.08	0.25	1.87
VA	8.16	254	3.95	−0.49	0.9912	0.034	0.03	0.09	1.29
pCA	9.35	254	0.83	−0.07	0.9941	0.028	0.10	0.33	1.03
FA	9.93	254	1.96	−0.02	0.9929	0.030	0.05	0.15	1.56
Naringin	11.26	280	2.51	−0.26	0.9978	0.016	0.02	0.06	0.36
Hesperidin	11.83	280	3.22	−0.31	0.9937	0.024	0.02	0.07	1.81
Quercetin	15.75	360	1.49	−0.99	0.9982	0.018	0.04	0.12	1.27
Naringenin	16.13	280	5.73	−0.34	0.9975	0.012	0.01	0.02	0.71
Hesperetin	18.11	280	8.56	−1.14	0.9978	0.014	0.01	0.02	1.64
Kaemferol	19.96	360	2.52	−1.38	0.9981	0.026	0.03	0.10	1.91

*t*_R_ = retention time.

**Table 2 molecules-29-02003-t002:** Precision and recovery of standard phytochemicals at 3 different concentrations.

Phytochemical	Intraday *n* = 3 (Precision [% RSD]; Recovery [%])			Interday *n* = 9 (Precision [% RSD]; Recovery [%])		
	0.5 µg/mL	2.5 µg/mL	5.0 µg/mL	0.5 µg/mL	2.5 µg/mL	5.0 µg/mL
Catechin	2.77 (88)	1.95 (91)	1.84 (98)	4.65 (84)	2.68 (88)	1.67 (94)
4-HBA	3.74 (84)	1.83 (89)	1.57 (97)	3.91 (86)	2.32 (87)	1.62 (95)
CA	2.73 (83)	2.54 (93)	1.62 (93)	2.43 (84)	3.64 (89)	1.54 (91)
VA	2.96 (89)	2.60 (95)	1.87 (96)	3.87 (83)	2.45 (90)	1.34 (97)
pCA	4.59 (85)	2.96 (95)	1.74 (93)	2.63 (89)	3.69 (90)	1.95 (93)
FA	3.91 (89)	1.64 (86)	1.64 (92)	2.86 (83)	3.45 (87)	1.63 (95)
Naringin	4.86 (91)	2.55 (86)	1.96 (90)	4.94 (82)	2.86 (91)	1.61 (97)
Hesperidin	3.82 (90)	1.54 (93)	1.89 (98)	3.32 (84)	3.75 (92)	1.39 (98)
Quercetin	4.98 (88)	1.74 (91)	1.53 (91)	3.43 (81)	3.89 (88)	1.54 (97)
Naringenin	3.93 (86)	1.57 (88)	1.68 (94)	3.35 (86)	2.88 (92)	1.58 (92)
Hesperetin	4.77 (83)	1.79 (92)	1.55 (96)	2.87 (89)	3.80 (87)	1.98 (95)
Kaempferol	3.51 (90)	1.92 (95)	1.32 (97)	4.43 (87)	3.88 (92)	1.49 (98)

**Table 3 molecules-29-02003-t003:** In silico ΔG (kcal/mol) results from enzymatic screening.

	Human Angiotensin- Converting Enzyme	Human Maltase- Glucoamylase	Human Sucrase- Isomaltase	Human Pancreatic Alpha-Amylase
Ligand	ΔG (kcal/mol)	ΔG (kcal/mol)	ΔG (kcal/mol)	ΔG (kcal/mol)
Lisinopril	−7.6	-	-	-
Acarbose	-	−7.9	-	−9.1
Kotalanol	-	-	−5.9	-
Amaranthin	−9.6	−7.1	−6.8	−8.9
Betanin	−9.5	−8.1	−8.6	−9.5
Catechin	−9.9	−8.6	−7.6	−9.5
Gomphrenin-I	−9.4	−7.8	−8.8	−8.6
Hesperetin	−8.1	−7.0	−6.5	−7.7
Isoamaranthin	−6.8	−9.1	−8.2	−11.9
Isobetanin	−9.7	−8.3	−8.0	−9.1
Isogomphrenin-I	−9.50	−7.4	−9.0	−8.4
Kaempferol	−8.0	−7.4	−7.7	−9.0
Naringenin	−8.1	−6.9	−6.3	−7.7
Quercetin	−8.3	−7.4	−7.8	−9.1

## Data Availability

Data are contained within the article.

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
