# Peer review of "HPLC-Based Metabolomic Analysis and Characterization of Amaranthus cruentus Leaf and Inflorescence Extracts for Their Antidiabetic and Antihypertensive Potential"

_molecules, 2024, doi:10.3390/molecules29092003_

Round 1
Reviewer 1 Report
Comments and Suggestions for Authors
The comments are as follows:
1. "Amaranthus cruentus L., also known as red amaranth, is one of the many species that has received attention for its exceptional nutritional profile and traditional medicinal use." Please, be more specific. What does exceptional mean in this context? What medical use?
2. "....and various pharmacological activities associated with amaranth that involve reducing oxidative stress". Please, specify which pharmacological activities.
3. Overall, the introduction section should be improved by inserting relevant data. It is too general. What is the novelty/originality of this work in comparison to the previously published data?
4. Line 74-81. This text does not belong to the introduction section. The authors should delete it or move it to the Conclusions.
5. Please use term "red amaranth" or "Amaranthus cruentus", not both.
6. Particle size and moisture content of dried sample material should be provided.
7. The authors should check the text for typos and grammatical errors. I suggest that you do not use the first person plural in the text.
8. Did all twelve authors make a significant contribution to this paper?
Author Response
Dear Reviewer, we are grateful for your valuable feedback on our manuscript. We appreciate them, and they have been great in improving the paper. We carefully reviewed the comments made and answered as best we could.
Comment 1: "Amaranthus cruentus L., also known as red amaranth, is one of the many species that has received attention for its exceptional nutritional profile and traditional medicinal use." Please, be more specific. What does exceptional mean in this context? What medical use?
Response 1: Thank you so much for your comments. We have made modifications to the text to display the nutritional values for amaranth, which are considered exceptional. The passage is now being displayed in the following way:
Amaranthus cruentus L., also known as red amaranth, is one of the many species that has received attention for its exceptional nutritional profile, which includes gluten-free, low calories, and low fat per serving, along with high-quality proteins and polyphenols, and for the amaranth-based meals used in traditional medicine [2].
Comment 2: "....and various pharmacological activities associated with amaranth that involve reducing oxidative stress". Please, specify which pharmacological activities.
Response 2: Thank you for your comment. We have included pharmacological activities in the text. The passage is now presented as follows.
“These compounds have been identified as the cause of the potential antioxidant properties and various pharmacological activities, including antidiabetic, antibacterial or antihelminthic, associated with amaranth that involve reducing oxidative stress [8-9]”
Comment 3: Overall, the introduction section should be improved by inserting relevant data. It is too general. What is the novelty/originality of this work in comparison to the previously published data?
Response 3: Thank you for your comment. The introduction section was made better.
Comment 4: Line 74-81. This text does not belong to the introduction section. The authors should delete it or move it to the Conclusions.
Response 4: Thank you for your comment. The text was moved to the conclusion section and the last two passages were preserved.
Comment 5: Please use term "red amaranth" or "Amaranthus cruentus", not both.
Response 5: Thank you very much for your comments. We decided to use A. cruentus in the text. The term “red amaranth” has been replaced by 'A. cruentus' and highlighted in green.
Comment 6: Particle size and moisture content of dried sample material should be provided.
Response 6: The particles in the stationary phase of the matrix solid-phase dispersion extraction were 40 μm in size. This feature can be found in section 3.2. Section 3.3 has been modified to clarify the use
of stationary phase C18. Lyophilized samples are the basis for all phytochemical quantification data, as mentioned in the main text.
Comment 7: The authors should check the text for typos and grammatical errors. I suggest that you do not use the first person plural in the text.
Response 7: Thank you for your comment. We have made many corrections to the main text.
Comment 8: Did all twelve authors make a significant contribution to this paper?
Response 8: The conception, design, achievement of the work, or analysis of the findings were all significant contributions made by all authors. Reviewing the contributions of each author is possible by looking at the 'Author Contributions' section.

Reviewer 2 Report
Comments and Suggestions for Authors
The article titled “Pharmacological Screening and Metabolomic Profile of Red Amaranth (Amaranthus cruentus L.): Evidence of a Functional Food” is an interesting and well-written manuscript focused on the investigation of the pharmacological properties of aqueous methanolic extracts of red amaranth.
The following points should be addressed before publication:
- The authors focused their analysis on the aqueous methanolic extracts from the leaves and inflorescences of red amaranth; however, the experimental procedure used to obtain the abovementioned extract is not reported throughout the manuscript. Please include a description of the extraction technique used along with appropriate references.
-While the HPLC analysis conducted by the authors has been accurately documented, it would be useful to add HPLC chromatograms of both extracts and standards to confirm that the analyte contents are correct.
-Page 7, line 243, the abbreviation “DPPH” should be spelled out.
-In Section 2.5.2 the authors performed in silico studies to evaluate the mechanism of action of phytochemicals on three distinct targets. In this regard, it is necessary to discuss and justify more in detail the rational behind the choice of the three human enzymes as targets. Additionally, it is imperative to address a concern regarding this section: hypothesizing a mechanism of action based solely on docking calculations of energy scores is inadequate. While a strong negative value in energy score theoretically indicates high binding affinity, it cannot stand alone without other experimental data. Therefore, in order to validate the putative mechanism of actions of the different compounds (e.g. as inhibitors of the human angiotensin converting enzyme), I recommend conducting competitive inhibitor binding assays for all ligands and performing re-docking or cross-docking studies. Moreover, improvements in the quality and presentation of docking images are advisable.
Author Response
Dear Reviewer, we are grateful for your valuable feedback on our manuscript. We appreciate them, and they have been great in improving the paper. We carefully reviewed the comments made and answered as best we could.
Comment 1: The authors focused their analysis on the aqueous methanolic extracts from the leaves and inflorescences of red amaranth; however, the experimental procedure used to obtain the abovementioned extract is not reported throughout the manuscript. Please include a description of the extraction technique used along with appropriate references.
Response 1: Thank you for your comment. We are sorry for missing this information. The preparation of the metabolites extract was as we previously reported in Araujo-León et al., 2023. We have made changes in the header title 3.3 and in the text to clarify the sample preparation.
Comment 2: While the HPLC analysis conducted by the authors has been accurately documented, it would be useful to add HPLC chromatograms of both extracts and standards to confirm that the analyte contents are correct.
Response 2: Thank you for your comment. The main text now includes chromatograms taken from leaves and inflorescences.
Comment 3: Page 7, line 243, the abbreviation “DPPH” should be spelled out.
Response 3: Thank you very much for the observation. DPPH now has the chemical compound named 2,2-diphenyl-1-picrylhydrazyl as an addition.
Comment 4: In Section 2.5.2 the authors performed in silico studies to evaluate the mechanism of action of phytochemicals on three distinct targets. In this regard, it is necessary to discuss and justify more in detail the rational behind the choice of the three human enzymes as targets. Additionally, it is imperative to address a concern regarding this section: hypothesizing a mechanism of action based solely on docking calculations of energy scores is inadequate. While a strong negative value in energy score theoretically indicates high binding affinity, it cannot stand alone without other experimental data. Therefore, in order to validate the putative mechanism of actions of the different compounds (e.g., as inhibitors of the human angiotensin-converting enzyme), I recommend conducting competitive inhibitor binding assays for all ligands and performing re-docking or cross-docking studies. Moreover, improvements in the quality and presentation of docking images are advisable.
Response 4: Thank you very much for your comments. The image was improved by increasing the resolution of the figure. In section 2.5, additional information has been added to support the choice of three enzymes that are crucial for treating diabetes and hypertension. Regarding the suggestion to perform cross-docking, this technique is not suitable since all protein structures already have a co-crystallized ligand reported as an inhibitor. A virtual screening of all compounds against the selected enzymes was conducted for this reason and according to the criteria mentioned. If we were capable of evaluating all the available structures and co-crystallizing them with all the available ligands, we would only be able to determine which amino acid residues are crucial for inhibitor ligands or agonists depending on the situation. Moreover, the crystal structures in the presence of different co-crystallized drugs show a different conformation or cavity since some ligands can be very small or of a different chemical nature. For this reason, not all crystal structures could be useful in this work, and we cannot predict compounds with a more extensive chemical structure than those of synthetic origin.

Round 2
Reviewer 2 Report
Comments and Suggestions for Authors
The present manuscript was revised by the authors according to my suggestions.
Author Response
Your feedback comments are much appreciated.